# Social Innovations for the Achievement of Competitive Agriculture and the Sustainable Development of Peripheral Rural Areas

Jadranka Deže *, Tihana Sudarić and Snježana Tolić

Department of Bioeconomy and Rural Development, Faculty of Agrobiotechnical Sciences Osijek, Josip Juraj Strossmayer University in Osijek, Vladimira Preloga 1, 31000 Osijek, Croatia; tsudaric@fazos.hr (T.S.); stolic@fazos.hr (S.T.)
* Correspondence: jdeze@fazos.hr

**Abstract:** The purpose of the study was to analyze social innovations in a broader context in order to achieve sustainable development goals. In terms of a response to the research questions, a comprehensive analysis of an alteration process in rural development was conducted while identifying the social innovations, highlighting the good practices, and conceptualizing the social innovation typology of peripheral rural areas. The methodology included a comparative analysis addressing three European regions, namely Northern, Central, and Western Europe, represented by Finland, Croatia, and France, respectively, together with their nine good practice examples during a biennial RUR'UP project period. The results demonstrate the typological differences between the selected examples of social innovations that stimulate the rural development of peripheral rural areas. These examples were formulated by the different social conditions in which these innovations were created. As a traditional economic activity in rural areas, agriculture is a great challenge for the application of innovations, can effectuate changes in the economic activities of the rural population, and can promote social and economic sustainability. In conclusion, significant differences among the regions were proven on the basis of social, environmental, and economic impacts arising from the application of social innovation.

**Keywords:** social innovations; sustainable development; agriculture; peripheral rural areas

## 1. Introduction

Progress in the fulfillment of the global sustainable development goals (SDGs) up to the year 2030 is not uniform, and many expectations have not been met. More than half of the period consisted of the transformation of national economies and the integration of the entire community, with goals including the social, economic, and environmental aspects of sustainability practices. The achievement of sustainable development through the seventeen United Nations (UN) SDGs is a priority development plan up to the year 2030 that was adopted by all the UN Member States in 2015. It is a joint plan considering the peace and prosperity of people, the planet, and the future of the world. The global goal is to eradicate poverty through development strategies that can double the income growth of small-sized food producers and family farms, as well as to stimulate economic growth in rural areas (SDG 2015) in the context of climate change, which represents an immediate danger and an additional challenge for both agriculture and peripheral rural areas.

The report on the results achieved in the past period (GSDR 2023) indicates the necessity of strengthening and revitalizing a global partnership for sustainable development (Goal 17). Simultaneously, in agriculture, investments are declining because fewer financial resources are being allocated to agriculture from state funds in relation to its contribution to the gross domestic product (GDP), which has declined globally—that is, from 0.50 in 2015 to 0.45 in 2021. All World Trade Organization (WTO) member countries adopted the

Ministerial Decision on Export Competition, thereby formally removing all forms of rights to agricultural export subsidies. Thus, the expenditure pertaining to the subsidies reduced from EUR 218 million in 2015 to almost zero in 2021 (Goal 2). Accordingly, it is necessary to investigate how much progress has been made while devising a process of sustainability for the development of agricultural and rural areas.

Peripheral rural areas in the European Union (EU) are covered by almost 30% of used agricultural farmlands and almost 15% of agricultural holdings (Keenleyside et al. 2014).

At the same time, they have the greatest importance for rural life, the preservation of traditional cultural heritage, biodiversity, and the preservation of ecosystems, such as carbon sequestration and water retention (see SDG 15: SDG 8—Decent Work and Economic Growth, SDG 15—Life on Land, and SDG 13—Climate Action). Agricultural lands of a high nature value (HNV), with areas occupying more than 25% of (EU) farmlands, contribute to the quality of agricultural products and to the preservation of biodiversity, cultural landscapes, territorial cohesion, and employment (Lomba et al. 2020).

These areas have great biological variety and a tradition of agricultural practices. They contribute to the development of the local economy because they have small production capacities adapted to geographical conditions, all of which generate great capital value. In contrast, long-term depopulation processes deplete human capital. Human potential constitutes an intellectual capital as a form of an intangible asset that becomes more important than a material one. The sustainability of farm operations depends on agri-entrepreneurs, with their knowledge, abilities, skills, and experiences. It is expected that they profitably and sustainably manage agricultural entities while adapting themselves to the rapidly changing environment (Deže et al. 2023). Social innovation in a rural area is, however, difficult to identify and measure; therefore, it is necessary to create methods for its categorization. One way of analyzing social innovation should, consequently, be based on the factors that are decisive for the emergence of social innovations and responsible for their sustainability (Neumeier 2012, 2017).

Compared to a metropolitan area, a rural area is perceived as a less economically developed zone characterized by traditional values, according to which development is slow or difficult and, hence, often leads to depopulation. In order to alter these trends, it is necessary to find new solutions and procedures that would render rural areas more attractive in social, economic, and ecosystem terms. Along these lines, both local and national government structures recognize the hidden potentials of underdeveloped peripheral rural areas when it comes to production and technology. Peripheral rural areas have special importance for each country's agriculture due to their natural and productional values, cultural and traditional heritage, and rural employment capacities. The introduction of innovations in such areas is particularly important in animal husbandry and in the cultivation of low-intensity arable land and permanent plantations due to the elevation of the level of productivity and profitability. The aforementioned facts contribute to the creation of assumptions aimed at the survival and sustenance of the local population and its progress with regard to rural development (BEPA 2014).

Social innovation represents an option that can solve social challenges while making use of government interventions. The interaction between social communities and public management demonstrates an increase in the interest in social innovation as a way of simultaneously creating social benefits and economic opportunities (Adams and Hess 2010). Bock (2012) connects the term "social innovation" with agricultural and rural innovations. In contrast, social innovations are rarely associated with agriculture as an economic activity, but are common in discussions about rural development. However, the needs of a society are identified in terms of the sustainability of production resources, the need for connections and cooperation, and a new knowledge streamlined because of the changes that are necessary for the revitalization of rural communities. In fact, social innovations are a set of interdependent processes and their beneficial outcomes. Marques et al. (2018) recognized the structural differences between social innovations that relate to a broader social change—that is, to an instrumental one, which uses the rebranding

of previous content in a way that is more attractive to participants, and targeted social innovations, which can be complementary to the socioeconomic conditions at present or can be radical in relation to them.

Innovation generally implies something new that is purposeful and represents certain social values. Social innovation refers to the development or significant improvement of strategies, concepts, ideas, products, or services with the aim to effectuate positive changes, in order to meet or to respond to social needs. They can contribute to a higher employment and consumption rate within a society and improve the quality of life (French et al. 2014). In urban areas, progress is more visible and easier to replicate, whereas the application of innovations in rural areas is a challenge, which can ultimately lead to multiple benefits (Bulimaga et al. 2022).

How radical should innovations be to cause radical changes? Klevmoen (2021) explains that there are major challenges associated with social, environmental, and economic changes in the development processes. Radical innovations are popularized as one of the answers to facing such challenges more effectively. They are applied in public discourse and are related to two dimensions—to wit novelty and change management. The novelty dimension possesses a category of radical innovation, and change management has several different categories that can be implied as the models of change process management, according to theoretical knowledge. Radical innovations are connected with disruptive changes in the products and processes. Such innovations are the basis of challenge management systems and their economic, social, and environmental dimensions. It is concluded that the main difference is in the approach and comprehension of radical innovations that are connected with the innovations in products, services, and processes. Simultaneously, it is necessary to monitor to what an extent the radical innovations may cause the changes in basic management systems and their economic, social, and environmental dimensions. The highlighted danger is, as Unger mentions (qtd. in Nicholls et al. 2015), that social innovation is absorbed into the existing systems: it is tamed into irrelevance and thus remains unnoticed.

Social innovations do not find their market in respect of ordinary customers until their quality meets the customers' expectations and customers' standards. Theory distinguishes radical innovation from what is called a "sustaining innovation." The development of rural areas is in this manner related to agriculture as the rural population's dominant economic activity, and innovations in agriculture are, therefore, the catalysts of socioeconomic rural development (Vercher et al. 2022; Wirth et al. 2023).

In the early 1990s, the concept of High Nature Value Farming: Learning, Innovation, and Knowledge (HNV-Link) was developed, with the aim of developing and implementing the innovations in agriculture and preserving natural values through a multi-collaborative approach in Europe. Peripheral rural areas (PRAs) are an important component of European agriculture because of their natural values and because of the high-quality agricultural products, cultural and traditional heritage, and rural employment. Agricultural land of high nature value (HNV) is described as an area on which agricultural activities are connected with exceptionally high biodiversity. Most frequently, agricultural production occurs in peripheral rural areas where there are natural limitations imposed on intensive production, mostly livestock farming with a low production intensity. The degradation of agricultural land, the abandonment of production, and degressive socioeconomic processes are the long-term threats to such extensive and naturally acceptable agricultural systems. It is a major challenge to increase the socioeconomic sustainability of high-nature-value agriculture while preserving the natural resources with the ecosystem services provided to the society. The HNV-Link network connects thirteen partners, being focused on the development and sharing of innovations that support the agricultural systems and social communities in the peripheral rural areas, while improving their socioeconomic sustainability and environmental efficiency through innovation. Thus, agricultural systems with high nature values are being created, and the innovations are being implemented therein. Such areas to which the innovations have been applied become the network's European learning

areas and serve to process the mutual learning and use of innovation expertise for the sake of an implementation in other regions with agricultural lands of a high nature value (HNV-Link 2019).

Agricultural lands featuring high nature values comprise European areas where agriculture is the main (or the most dominant) land use and where that type of agriculture supports, or is linked to, a high diversity of species and habitats, the presence of species of Europe's special conservation interest, or to both (Andersen et al. 2007).

Koloszko-Chomentowska and Sieczko (2018) highlighted the importance of a non-market function of agriculture—namely, the sustainability of socioeconomic activities in the sparsely populated areas, the protection of natural environment, and the preservation of rural cultural heritage. The realization of these functions is connected with the policy of economic diversification in rural areas. The research results confirm the justification of the maintenance of a leading role of agriculture; therefore, other economic sectors in the region should support its growth. At the same time, the share of rural areas is increasing, whereby the conditions are created for tourism industry development and the activities that popularize a healthier way of life due to preserved natural resources.

Judging from the aforementioned facts, agricultural management and rural area development are obviously under the influence of social innovation application, which is indeed a complex process. This progression precisely provides a discursive strength through the connection between an agrarian policy and sustainable development, as well as through a latent weakness that arises from the burden of social innovation role.

Ascani and De Vivo (2020) concluded that, in the political documents of the European Commission (EC), the most attention is paid to social issues and innovations, the contribution to climate change, environmental goals, risk management, and sustainable production. It is directed to the new forms of supply chains, such as bioeconomy, and not to diversification activities, such as social farming. The innovations are to be found in the context of employment and economic development. The importance of a territorial dimension of development policies is emphasized because the goals of Europe 2020 are not sufficiently adapted to the specificities of regional and local communities, and a territorial differentiation is, therefore, necessary. The alternative models of "other small-scale agriculture" that contribute to the construction of the "second welfare" are a key to the creation of innovation and social cohesion. The way toward enacting these processes is in reinforcing the socioeconomic fabric of rural areas. Navarro-Valverde et al. (2022) stated that the experience of implementing the LEADER approach results in significant cultural and social changes. The perceptions and perspectives of the local community are the basis of one's own path toward development through a greater awareness of local resources. De Pieri and Teasdale (2021) observed two groups of policy instruments designed to facilitate the application of social innovations because they are important for development processes. The first was democratic innovations—namely, private financing for social welfare needs encouraged through social and public investments. The second was focused on political instruments for the application of social innovations beyond the public sector through four stages of social innovation development. In the phase of crisis—that is, on the occasion of a status quo—the policymakers enable an interaction between the unrelated groups to generate new ideas and innovations. This is followed by the processes of the evaluation, selection, and implementation of successful options. The subsequent stage is exploitation, when the resources are used and barriers are removed; however, the policies are then necessary to enable a demand creation and market development. In fact, social innovation requires political approaches that secure the resources and increase resistance to future changes.

The following research issues were formulated in a search for social innovation contribution to sustainable development—that is, to each of its pillars. The first question referred to the research of whether environmental impacts on a social innovations system in the PRAs (RQ1) are evident.

The scientific sources encourage curiosity concerning the forms of environmental impacts in the application of social innovations at present. According to Kontula and Raunio (2019), the area of traditional rural habitats has decreased by more than 90% since the 1960s, as they have been replaced by cultivated grasslands and arable crop areas that mainly feature crop rotation. At the same time, grazing in forests, coastal areas, and wetlands became rare. Due to such a dramatic decline in the representation of seminatural habitats that have become the most threatened areas, the importance of managing the remaining areas is emphasized in the research. The importance of biodiversity in traditional rural habitats is thus highlighted by Hyvärinen et al. (2019), who claimed that one quarter of endangered species and forty percent of all extant species live in such habitats. In the follow-up research, Mäkeläinen et al. (2019) supported the claim while including numerous species of native flowers, fungi, butterflies, moths, and birds. The innovation was created due to a need to maintain traditional rural habitats, which include seminatural grasslands, wooded meadows, and pastures. Grazing management problems arose due to the specialization of farms. Many of them had seminatural pastures, lacked livestock, or the farmers possessed livestock but were not the pasture proprietors. The establishment of services for searching and connecting the pastures' owners with potential users and livestock owners facilitated the cooperation and long-term contractual relations—that is, they provided the guidelines for cooperation, the templates of various forms of contracts, and the division of costs and contractor responsibilities. Partanen's (2022) research examined the role of social innovations in the development of the tourism industry through a framework that emphasized local cooperation and joint creation to enhance local resilience. By virtue of his research, it was concluded that social innovation is used to build resilience and potentially improve the sustainability of development through tourism. The solutions to the challenges were detected while increasing diversification, local culture inclusion, multisectoral cooperation, and an environmental approach in order to achieve long-term resilience. How to use rural resources was explained by Panapanaan (2010) by virtue of an innovative approach that identified a perspective countryside while exhibiting competitiveness and benefits for a community. The research conducted by Lehtimäki et al. (2019) focused on environmental and innovative approaches to food production practices through social entrepreneurship.

The second question was related to seeking out the economic effects impacting the social innovation system in PRAs (RQ2).

The changes affecting farmlands in their transformation of a relationship toward ownership and the application of social innovations were monitored through a framework of scientific sources. It was established that social innovations are prospectively applied to the alterations in a relation to the property and farmland use. Accordingly, there was evidence of a mutual connection between social innovations and a modification in the mechanisms of a property–use relationship (Léger-Bosch et al. 2020). The development by virtue of community-supported agriculture in France (*Associations pour le Maintien de l'Agriculture Paysanne*, AMAP) and Austria (*Community-Supported Agriculture*, CSA) was investigated in the context of social innovation. That form of community-supported agriculture is not only applied in practice, but it also has formal territorial organizations that provide support to the farmers. The experiences confirmed that the farmers and their needs played a central role in rural development practices as a driving force on an operational level. Thus, the establishment of the cooperation between the AMAP organizations and organic farmers has a significant role. What can serve as an example of good practice was the importance of forming and strengthening alliances among the established players who can render technical support on an operational level in the agricultural sector (Egartner et al. 2020). A connection of these effects was created as a result of the joint efforts of researchers, policymakers, strategic actors, and innovation brokers in agrifood systems. This situation was a result of following a road map toward sustainability transitions in agriculture. Such major changes included the coevolution of a multitude of related elements, technology, infrastructure, management, scientific knowledge, industry, and

other connected institutions, linked by a common endowment called System Innovation for Sustainable Agriculture, or SISA (Barbier and Elzen 2020).

The third was linked to the question of whether social impacts on a social innovations system in the PRAs (RQ3) existed.

According to the scientific sources, the responsibility of agricultural stakeholders is to face the social and climatic challenges faced by an ecosystem while applying innovations based on digital technologies and new concepts. Their key role is to aspire toward economic, environmental, and social sustainability in the agricultural sector (Hrustek et al. 2022; Čehić Marić et al. 2023). In that context, in Croatia, social innovations are closely related to social entrepreneurship and social enterprises. "Innovation-driven social enterprises," as one of the more explicated types, were based on an innovative solution to the social problems while combining the existent resources to produce a new value and common benefits for a wider social community (Vidović and Baturina 2021). The form of social entrepreneurship was formalized through a legal basis on a national level. Despite this, support for public policies in the encouragement of the development of social enterprises is insufficient. A model for the achievement of change consists of an innovative approach to a solution to economic and social problems. The business sector is encouraged to become more involved in social programs through cofinancing, ensuring access to the market and mentorship of nongovernmental organizations, or NGOs (Tišma et al. 2022). The introduction of social innovation is significantly related to the probability of growth pertaining to the success and internationalization of businesses and small- and medium-sized enterprises, or SMEs (Begonja et al. 2016). As a result, social innovation and social entrepreneurship may become tools for the diversification of tourism and hospitality industries' products, and thereby contribute to environmental, economic, and social sustainability practices (Matošević Radić et al. 2020). The operation of pop-up rural social innovation hubs enables the connection of existing activities, as well as an upgrade to new ones, in which sustainable development is achieved. Entrepreneurial mentality, ethics, and a tendency to introduce innovations represent a social capital that should be directed to the creation of business ideas and development plans of local rural hubs. The strengthening of rural local communities is achieved by initiating socio-innovative, sustainable microentrepreneurial activities. In this way, rural hubs become the very flexible models that not only contribute to sustainability, but also welfare, a better lifestyle, and a safer existence in rural communities (Kantar and Svržnjak 2019).

According to the Social Innovation Policy Framework (OECD 2016) that was created in accordance with the EU's 2020 strategy, the focus was placed on smart, sustainable, and inclusive growth, which is relevant to social innovation. The initial phase of implementation began in 2016 with a request for funds from the national budget, aimed at raising awareness of social innovations and creating better cooperation mechanisms for the purpose of understanding the needs on local levels.

The final question was to understand the differences in social innovations that exist between European countries (RQ4).

The differences were created due to the impact of national welfare and economic development programs on history, which has exerted an effect on social innovation initiatives and forms of collaborative governance (Merlin-Brogniart et al. 2022). Support for social innovation and local actors is crucial for their efficiency in the face of external challenges and opportunities. Strategic thinking about the key dimensions aimed at the design of social innovation activities in marginalized rural areas should focus on the management of social networks, financial sustainability, and know-how (Marini Govigli et al. 2020).

In accordance with the previously stated facts, the main aim of this study is to recognize the role and types of social innovations in sustainable agriculture and the development of peripheral rural areas, and to achieve the appropriate effects through social, environmental, and economic aspects. Thus, the paper applies a comparative analysis to determine the relationship between social innovation effects on sustainable agricultural and rural development areas. We analyze the current processes of social innovations in some European

countries while monitoring the activities of different actors in public and private sectors. In some fields, the potential for agribusiness is identified when it concerns the investors and other stakeholders interested in social welfare.

## 2. Methodology

For almost a decade, social and economic development have been striving to achieve the relevant SDGs, whereby the latent potentials have been precisely found in rural areas and agriculture as a traditional economic activity. Many rural areas, however, are gradually lagging behind, which is caused by the depopulation processes, while urban areas are developing and experiencing problems associated with overcrowding. In such a situation, a constantly increasing gap is being created. The application of social innovations is a useful tool for healing wounds and mitigating the consequences of recent decades. In order to monitor the effects of social innovations on sustainable development, the following research questions were defined:

RQ1: What are the environmental impacts on a social innovation system in the PRAs?
RQ2: What are the economic impacts on a social innovation system in the PRAs?
RQ3: What are the social impacts on a social innovation system in the PRAs?
RQ4: What are the differences in a social innovations system between the European countries?

Therefore, the main sources for the collection of the material used in the analysis were as follows: scientific papers; the materials of local, national, and international institutions; and the written findings pertaining to the participants from seven partner countries in the RUR'UP project, entitled *Innovative Education for Sustainable Development of Peripheral Rural Areas* within the Erasmus+ 2020 Program. The participants were from Bulgaria, Finland, France, Greece, Ireland, Romania, and Croatia. Thus, the data used for this paper's analysis were a product of a quadrennial study whereby two continuous international European projects were conducted, within which the Virtual Case Study Bank was created.

The collection of data used in the research was initiated by the HNV-Link project activities (HNV-Link) in ten areas (Bulgaria, Croatia, France, Greece, Ireland, Portugal, Romania, Spain, Sweden, and the UK, respectively), domineered by a high nature value of farming systems in which the innovations were applied. Agricultural lands of a high nature value (HNV) are those where agricultural activities support biodiversity and are also valuable in terms of cultural heritage, high-quality products, and rural employment. Most frequently, however, they have accommodated the productions of a low intensity—for example, extensive animal husbandry. At the same time, they are protected as the Natura 2000 habitats (by the EU legislation and as National Parks). They have an exceptional recreational value as well, and are the focus of rural and natural tourism industry's development practices. The HNV-Link, that is, High Nature Value Farming: Learning, Innovation, and Knowledge, collects and shares the innovations that support agricultural systems and communities in order to improve socioeconomic sustainability and environmental efficiency. By virtue of joint partner activities, such areas were recognized and became the learning areas (LAs) for mutual learning on an innovation level while using the experience acquired in other regions concerning agricultural lands of a high nature value. The goal was to contribute to the sustainability of agricultural areas of a high nature value while building capacity for innovation as well as learning how to successfully implement it. Through the LA network, specific needs and problems, as well as the innovations that were the best solutions to the problems and case studies, were identified in close cooperation with local actors. The LA representatives presented their innovations at the Innovation Fair and selected the cases opined as being the best suited to their needs (Bernard et al. 2023). According to Gaki et al. (2022), subsequent to the completion of project activities, the RUR'UP project (2020–2022) built on the activities performed within the HNV-Link project—namely, the *Innovative Education for Sustainable Development of Peripheral Rural Areas*, within the Erasmus+ 2020 program. The Customizable Personal Development (CPD) platform, entitled Teaching and Learning Online Courses: Dashboard (CPD 2022), collected innovative solutions from

the previous HNV-Link project dedicated to learning about contemporary challenges confronting sustainable development practices. There were the benefits derived from the RUR'UP project activities that identified the educational needs for sustainable development in the EU's peripheral rural areas; therefore, a case study bank was created to serve for the learning-from-example purposes, while universities were offered an e-course distributed via an online learning platform. All cases shared the same methodologies: an analysis of the key points of a baseline assessment, a mind mapping tool, the organization and conduct of an efficient farming demonstration event, the identification of vulnerabilities and adaptive capacities of a territory, stakeholder analysis in a participatory innovation process, an introduction to the innovation types, and the development of participatory projects in peripheral rural areas. The Electronic Platform for Adult Learning in Europe (EPALE 2023), storing the entire Virtual Case Study Bank, was deployed for a wider public distribution. From the case study collections/library for each country, the respective case studies were selected in accordance with the research objective and planned methodology. It was these databases and materials that formed the foundation for the further analyses conducted. A map of Finland, France, and Croatia depicts a territorial case distribution (Figure 1).

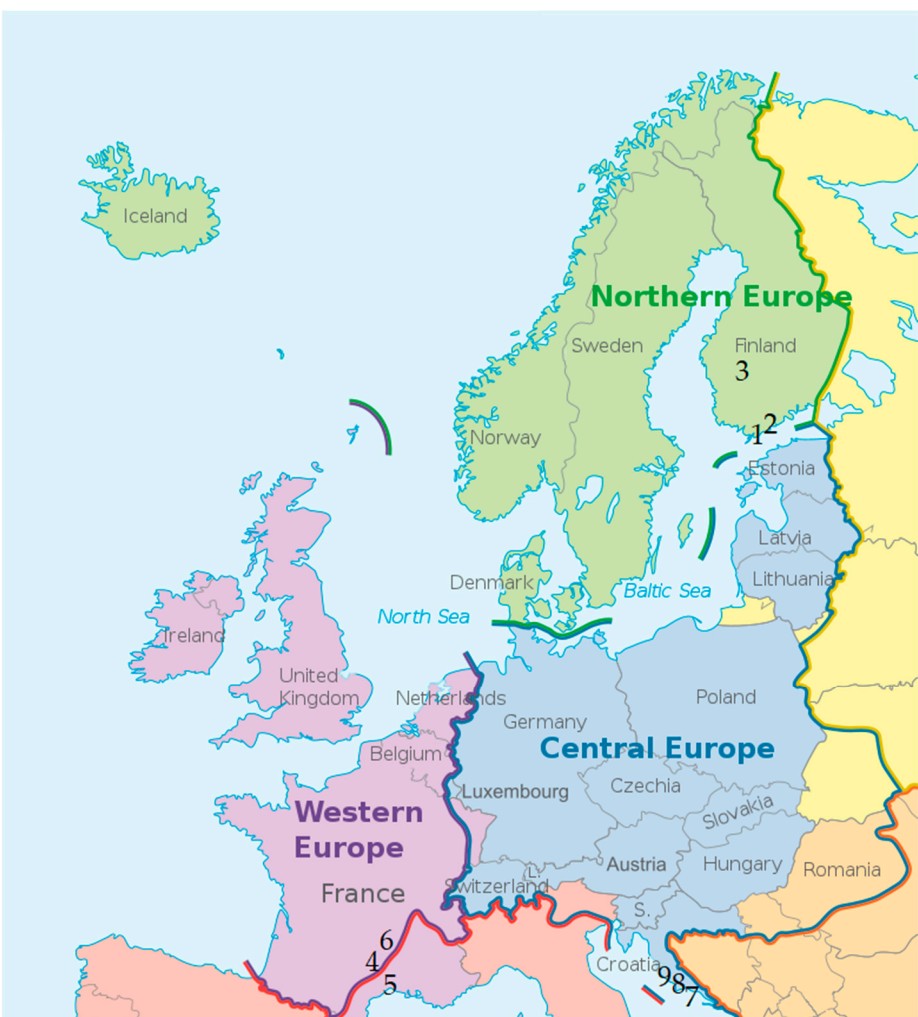

**Figure 1.** A map of the territorial distribution of cases in Northern, Central, and Western Europe. Source: authorially designed according to Jordan (2005).

The innovative processes that have contributed to the changes in peripheral rural areas in a long-term manner were the cases selected for a comparison of social innovations among the Northern European area (NESI; 1-3) exemplified by Finland, the Western European area

(WESI; 5-6) exemplified by France, and the Central European area (CESI; 7-9) exemplified by Croatia. The data for the analyses of nine cases were collected from the selected locations representing the peripheral rural areas on the national levels of the three selected countries. The selection of the cases was based on the following criteria:

a.　　The occurrence of an HNV agroecosystem defined by its land cover and farming systems according to the HNV farming typology (Andersen et al. 2007);

b.　　The existence of agroecological management instruments on a territorial level (e.g., the Natura 2000 area, a UNESCO site, National Park, a locally led agroenvironmental scheme under the Common Agricultural Policy's (CAP's) rural development programs, etc.); and

c.　　the presence of a multi-actor cluster (e.g., private entrepreneurs, professional/farming organizations, local authorities, universities and/or research centers, and NGOs) willing to be engaged in agroenvironmental management to support the HNV areas.

Those cases belong to the learning areas covered by the European countries (Bernard et al. 2023).

The research was focused on the cases located in peripheral rural areas whose activities contributed to economic, environmental, and social changes. At the same time, it was assumed that they were the catalysts of economic development on a local level. A selection of the cases analyzed considered the fact that they represented a local population that recognized its social needs, undertook local initiatives to face the social development challenges, and created local innovations that changed the traditional ways of acting.

A method to analyze the impact of social innovations on sustainable development and a categorization of social innovations was developed according to Rüede and Lurtz (2012), whose systematic conceptual literature review determined the types related to different comprehensions of the concept of social innovation. The indicators were clustered into coherent groups that represented a mutual context.

Furthermore, social innovation plays a crucial role in sustainable development, which is recognized by its social impacts. In European countries, the activities of national institutions have a direct impact on the structure of a social innovation system. The scientific research and practical cases confirm that different social innovation systems are being developed, specifically the Anglo-Saxon, Continental, and Eastern European types. By identifying the aspects of social innovation systems, the types that present common characteristics can be more clearly observed in the literature (OECD 2016).

Based on similar methodological procedures, the collected data were categorized and grouped. The determination and separation of indicators (15) of the social innovation system were conducted according to their types, which divided them into three pillars—that is, social (5), environmental (5), and economic (5)—with regard to their impacts on sustainable development. Comparative research results were processed and used to identify the different types of social innovation systems according to the respective countries.

A case analysis method was applied using a smaller number of samples and focus cases, in which the impact of social innovations in the peripheral rural areas of selected geographical zones was described. The authors analyzed the social innovation cases while reading the materials created by project activities (e.g., HNV-Link and RUR'UP), whereby the authors also created the codes for individual occurrences of the key impact on social innovations detected in the text. In the analysis continuation, the individual codes were linked to the categories representing the individual social innovation types. The five indicators that appeared most frequently according to the types were selected. The qualitative data pertaining to the impact were qualified according to the types of social, environmental, and economic social innovation systems. In order to contribute to the credibility, the knowledge about the cases was collected until the moment at which no additional, important, or new facts emerged to become the impact indicators. This was followed by the classification of indicators, according to the occurrence of impacts (social, ecological, and economic) in the analyzed geographical areas (NESI, WESI, and CESI).

In addition to the previous technique, the analytical technique illustrated below was also applied to the research. Qualitative research was conducted and the experiences acquired by the social innovation application were observed.

Thus, an analytical process–activity framework was constituted:

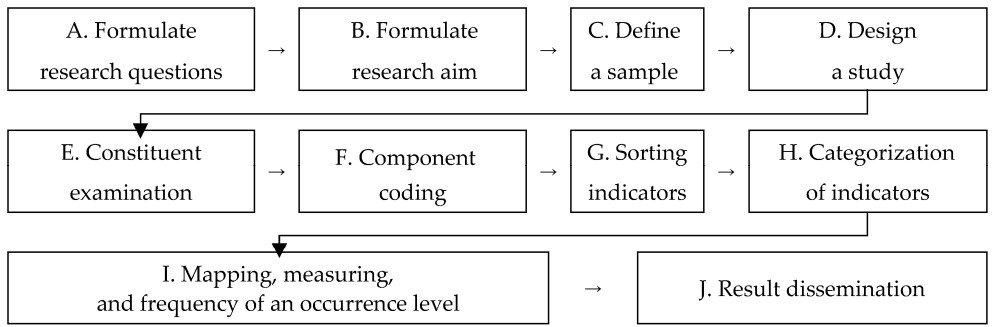

During a review of the relevant and recent scientific research, research questions (A) were formulated, resulting in a general research aim (B). The purpose of the analysis was to profoundly investigate the case contents and to acquire new in-depth knowledge of each individual case. A distribution according to the zones was performed in accordance with the planned methodology, while case selection was based on the availability of these cases on the platforms and in the databases implemented (C). An analytical process framework was construed, created in accordance with the research questions and a general research aim (D). Qualitative analysis applied descriptive iterations and a systematic compilation of a "key-term" list, attempting to emphasize the sense of the collected data while creatively facilitating the insights into each of them (Gibson and Brown 2009). Measuring instruments were modified and the data were aggregated up to the moment of a "saturation point"—that is, up to the moment where the collection of additional data obtained new information, or "theoretical sampling" no longer occurred (Pavić and Šundalić 2021). In accordance with the previous case, the texts were perused, while the textual key terms were marked by different colors, having thus become the coding process constituents. Therefore, a constituent structure was created, based on the description of the content and on the investigated case elements (E). This was followed by the action of encoding pertaining to the constitutive components identified in the process of a qualitative examination in order to use the short phrases to continue the analysis for the purpose of a symbolical highlight of the essence of a phenomenon. In such a way, the action of encoding was also performed as a part of the linguistic and visual metadata. According to Saldaña (2021), the aforementioned action employs an analytical technique relying on an attribute-based encoding method, thus serving as a data management practice but also as a process of abstraction and interpretation. The action of encoding was consequently performed in accordance with the nature of the components and research objectives (F). Qualitative data were organized by encoding, which was an important feature of an analytical process function, enabling rigorous data verification. It served to connect the different data segments. By combining the data segments, data series categories sharing common properties were formed (Coffey and Atkinson 1996). The main encoding objective was to facilitate the detection of data segments that could be separated and categorized in the same group. In that way, the indicators were also drafted, fulfilling a goal of the creation of a narrative that represented the original data. Hence, the indicator sorting process prepared the data for an evaluation process in all segments, and the indicators (15) were identified, representing the groups of data necessary to perform a categorization procedure (G). An analytical step collected the data that were identified as relevant to the research goal. They subsequently served as the tools for the generation of ideas on how to conduct an analysis thoroughly and precisely, and established connections between locations in the data and in the sets of data; therefore, in that sense, they were heuristic tools. The aforementioned datasets implemented a process of merging the segments into categories (Coffey and Atkinson

1996). As an integral part of the encoding cycle, indicator categorization was performed with the same purpose as a sorting activity—that is, as a preparation for a frequency measurement procedure. The indicators, however, were generated in accordance with the three pillars of sustainable development, implying the social, environmental, and economic impacts exerted on the sustainable development of peripheral rural areas (H). By aggregating the data on the basis of sorting and categorization steps, an analytical process of frequency mapping and the measurement of different data occurrence levels could be accessibly initiated (Coffey and Atkinson 1996). A simple mapping of indicators according to the researched zones (i.e., NESI, WESI, and CESI, respectively) was applied through the expression of their effectiveness in a local community with regard to the types (i.e., social, environmental, and economic impacts). A measurement of efficacy was based on the explicitly stated stakeholders' effects that became evident in the scope of social, economic, and environmental impacts on the analyzed cases. The effectiveness of a social innovation application recognized by local communities was marked by a positive symbol (✓), while everything else was marked by an (X) symbol. The level of occurrence of each individual indicator was expressed by the ratio of partial presence in relation to a total set, such as frequency (1/3, 2/3, and 3/3) (I). The results of the conducted research are related to the research questions, on the basis of which the paper contributes to the comprehension of social innovations in the development of agricultural competitiveness and the sustainability of peripheral rural areas. (J) Although an ethical codex is usually necessary in the case of qualitative research, this was not the case in this study because it involved the deployment of publicly circulated databases (Virtual Case Study Bank).

## 3. Results

There is an increasing interest in the research in the forms of innovation that create social benefits and new economic opportunities, which is also confirmed by the trends in change management processes. An analysis of the European development trends in peripheral rural areas proves that there are enough experiences on the basis of which one can understand what social innovations are and what their importance is for development policies. Social innovation practices are identified, and it is shown how they can be theorized and linked to applicable forms. The introduction of innovations in rural development is still unclear to the practitioners and scientists alike; therefore, the paper attempts to conceptualize the differences between the types that have an increasingly wide social application. It is necessary to enable sustainable rural development in peripheral rural areas. Achieving such a demanding goal is obtained by the comprehension of the social, environmental, and economic characteristics of an area. Social innovations can be one of the solutions to the challenges of achieving sustainable rural development, which is attained through both the recognition of needs in local communities and through the recognition of opportunities for sustainable social and economic development practices. By analyzing the examples of innovation processes on an international level, good entrepreneurial practices that have contributed to the economic empowerment and revitalization of rural areas can be observed. Thus, the paper analyzed the practices of applying social innovations through the cases of Northern European countries (NESI) while implementing the examples of Finland, Western European countries (WESI) exemplified by France, and Central European countries (CESI) using the example of Croatia, as shown in Table 1:

**Table 1.** A practice of innovative sustainable development in peripheral rural areas (PRAs).

| Examples | Innovations in the PRAs |
|---|---|
| NESI | NESI[1] Southern Finland: Association of Natural Pasture Importers.<br>NESI[2] Finland/Baltic Sea Catchment Area: E-College for Regenerative Farming.<br>NESI[3] Southern Finland: Pasture Bank. |
| WESI | WESI[4] Southern Massif Central: A Collective Approach for a Municipal Pastoral Pact.<br>WESI[5] Languedoc–Roussillon: Collective Approaches by Breeders–Official Labels Identifying Quality and Origin.<br>WESI[6] Causses and Cevennes: The Development of Direct Distribution to Farm Stores. |
| CESI | CESI[7] Island of Murter: Action Plan for the Sustainable Use of Resources.<br>CESI[8] Dalmatian Islands: Multistakeholder Organizations Fostering the HNV Products and Practices.<br>CESI[9] Dalmatian Islands: HNV Farming as a Tourist Activity. |

*3.1. Northern European Social Innovations in the Peripheral Rural Areas*

NESI[1]—Southern Finland: Association of Importers of Natural Pastures (Luonnon-laidunliha tuottajat ry, NPMPA 2022) has promoted the producers and protected the endangered traditional habitat in Finland's rural areas through innovative activities since 2020.

The purpose of the association is to unite the meat producers from natural pastures in Finland and to strengthen the production of environmentally friendly meat in accordance with animal welfare standards. It mainly concerns the extensive production and breeding of sheep. One of the goals is to expand the knowledge of customer relationship management (CRM) in order to increase the profitability of farmers and the demand for naturally pastured sheep meat in the market.

In the long term, it is expected to increase the utilization of natural grasslands, which are also restored by livestock grazing. Such a nature–pasture–meat concept is the basis for demand growth.

The term "seminatural grasslands" refers to the areas that have traditional habitats and are not cultivated and fertilized. A campaign plans to increase the visibility of meat from natural pastures and create special certificates. The major challenge is in conducting innovative activities aimed at the profitability of farms that use seminatural grasslands in the production process. By increasing the visibility of meat obtained from natural pastures, it becomes more attractive to customers, increases demand, and creates added value to the products. A higher level of added value is created by finalization through product variations and product diversification. The challenge of innovating in this form of association is found in both the connection and support between farmers and a solution to the processes of biodiversity loss.

NESI[2]—Finland/Baltic Sea catchment area: E-College for Regenerative Farming. In Finland, an innovative agricultural E-College for Renewal of Agriculture (E-College 2022) was established, with the aim of educating individuals about farm management, improving profitability, and protecting the environment. E-College aims to regenerate the farms on PRAs. The lectures are delivered in Finnish and Swedish and intended for 5000 people: farmers, students of agriculture, and others who may be interested in the subject.

The E-college was advertised in different ways through the media; however, since it was created through the cooperation of many organizations and partners, it was also presented in many other ways of advertising. The E-College is one of the start-up projects under the Carbon Action, which included the UN Climate meeting in the UK in November 2021. Carbon Action organized many events and webinars in support of its projects, which were mostly aimed at participating farmers. This included field days that were held several times a year and included practical training held by the Advisory Sector and scientific institutions. The topics included were agroforestry and the usage of manure.

The target of the Carbon Action Club was to initiate the launch of Pilot Carbon Farms that were linked to the research projects. All members of the Carbon Action Club

have access to events, webinars, and a Facebook group for connections and discussions. Translations into other languages and the adaptation of the material to different farm systems, soil types, and biophysical conditions is recommended. This will also, in the future, result in a collection of farmers' experiences with their practices in farm regeneration.

NESI[3]—Southern Finland: Pasture Bank (Laidun Pankki 2022) is a service portal that brings together the keepers of grazing animals, owners of grazing pastures, and farmers who raise livestock. It was founded in 2005 by ProAgria (2022), an association of local centers for the provision of services and knowledge in agriculture and rural enterprises, the Rural Women's Advisory Organization (*Majaa Kataluusmaiset*), and Agrifood Research Finland (*Majaa Elintarvikealojen Tutkimuskeskus*, MTT).

Seminatural grasslands and other natural habitats that have the greatest biodiversity are considered lands of high nature value (HNV) and are called traditional rural habitats in Finland, which have been created by traditional grazing and forage collection.

*3.2. Western European Social Innovations in the Peripheral Rural Areas*

WESI[4]—Southern Massif Central: Collective Approach for an Inter-Municipal Pastoral Pact (Causses Aigoual Cévennes 2022); the French partners implemented the activities that encourage innovation in the sustainable development of peripheral rural areas. In order to raise awareness of the specifics of the area and its potential for agricultural production (Agropastoral) in the Causses and Cevennes area, a large number of communication and dissemination activities were conducted (e.g., a project brochure, presentations, posters, on-site collective workshops, website publications, and a face-to-face workshop). Such activities identified the opportunities, challenges, and threats, as well as the activities that needed to be implemented in order to ensure the sustainability of development. Based on the close cooperation of the leading tripartite partnership (Triple Helix model of innovation) and a participatory approach, a cooperation and development strategy was created that included all actors (stakeholder mapping) and priorities, namely (1) to gather local stakeholders who have livestock production and promote product exchange and joint development in that area, then (2) exchange experiences and knowledge, including the specifics of the area, and (3) promote the needs and proposals of the local rural area. The process of exchanging information between the stakeholders occurred on cross-visits, with the aim of learning about innovations and recognizing the possibility of transferring innovations. The French partners created a matrix with innovative experiences in application and innovation needs identified in other areas. In the matrix, experiences and needs were matched, and partners agreed on activities and developed learning programs for field visits. In connection with the agreement, the Causses and Cévennes delegation participated in the cross-visit activity in two different ways: as a visitor, visiting the Burren area in Ireland, and as a host, hosting in the Causses and Cevennes area by two delegations from the area: Pindos Mountains (Greece) and La Vera (Extremadura, Spain) (Causses et Cévennes Regional Meeting Report 2018).

In the summary of the French Innovation Seminar (Girardin et al. 2017), the participants cited the innovative approach of the Pastoral Pact concerning land management or the European Life + Mil'Ouv Program for the conservation of pasture areas. However, the actors highlighted the important barriers and threats to the development of pasture activities: a lack of access to the land, an unattractive shepherding profession, a loss of the concept of multifunctionality of livestock, an ignorance of the impact of climate change on pastures and forests, and a loss of collective thinking and attitudes.

WESI[5]—Languedoc–Roussillon: Collective approaches by breeders–official labels identifying quality and origin: protected designation of origin (PDO), protected geographical indication (PGI), and traditional specialties guaranteed (TSG). Faced with a need to improve the products and recognize the practices in production technology, French farmers joined forces to improve the sales of their products. The goal was to achieve the value added pertaining to the farm-produced product. Through a collective approach, the growers created official marks that identified the quality and origin and brands that existed in

the region. At the same time, consumers perceived these labels as a guarantee of quality; therefore, the labels became increasingly popular among consumers who preferred the products that were geographically and traditionally connected to their area. Through such products, profitability on farms has improved. Innovation is related to the integration of the use of local resources and NVU in specifying product quality, especially when it comes to livestock products. The general meaning of such collective activities is to empower farmers and undertake actions that are usually too complex for an individual. Obtaining PDO, PGI, and TSG marks is a long and demanding process of preparing documentation, which is conducted much more easily by virtue of a collective endeavor, provided that the collective is connected to the regional institutions and civil society (HNV-Link 2020).

WESI[6]—Causses and Cevennes: The development of direct distribution to farm shops and Agrilocal (CENLR 2022). The development of direct distribution has different forms of distribution points: those that have been recognized in France for several years, and those that have not yet been recognized and need to be further promoted. The farmers' markets enable a joint distribution, which is increasingly popular with consumers searching for a direct connection to producers. In recent years, numerous agricultural stores were opened in the Causses and Cevennes area, and the Agrilocal platform was created to connect and network them. This platform brings together the suppliers of local products (farmers, processors, and local traders), caterers, and other institutions (schools and retirement homes) and in a simple, direct way while complying to the public procurement code, ensuring the delivery of products. The development of direct distribution improved the economic autonomy of producers. At the same time, it affected the sensibility of consumers toward local food and local producers. It is realistic to expect the risks that arise as a result of misunderstandings between the stakeholders in the network platform that can lead to a self-initiated exclusion or expulsion from the collective, as well as the deficiencies of self-financing practices.

### 3.3. Central European Social Innovations in the Peripheral Rural Areas

CESI[7]—Island of Murter: Action Plan for the Sustainable Use of Resources (Argonauta 2022) is an example of innovation in rural development conducted pursuant to the Action Plan for the Sustainable Use of Resources in the Area of the Island of Murter, Lake Vrana, and the Kornati National Park. The changes in the conversion of traditional to modern HNV practices were managed with the aim of transferring the knowledge about traditional practices that are disappearing due to the abandonment of agriculture and the aging of the population in rural areas. The local community's knowledge and experiences were revived and activated according to the guidelines on nature protection and the sustainable use of its resources. The concept of sustainable tourism was created through the synergy of nature protection and the development of Murter Island. A platform involving the islanders was established to strengthen the capacity of stakeholders and develop a participative management of the local area. It is an example of the practice of the participatory management of the protected areas modeled on French eco-museums that revive the local traditional practices of the sustainable use of natural resources (Botica et al. 2017).

CESI[8]—Dalmatian Islands: Multistakeholder organizations fostering the HNV products and practices—in the Dalmatian Islands' area, there are several LEADER organizations that indirectly promote the HNV (LAG-5 2022). By cooperating with the public, private, and civil sectors, they establish a form of cooperation in order to obtain EU funds for local projects that promote the rural development and sustainability of the agroecosystem. An example thereof is the Slow Food Pelješac Convivium, which promotes both branded local products and the use of food based on the principles of good, clean, and fair consumption practices. The activities focus on the promotion of seasonal food products, the protection of biological diversity, local recipes, foodstuffs, and the food that has been forgotten or is in some way endangered. At the same time, the producers and consumers are influenced through the promotion of organic agriculture without the use of pesticides (Andlar et al. 2017).

CESI[9]—Dalmatian Islands: HNV farming as a tourist activity. According to Abdessater et al. (2017), the examples of innovation bring together different stakeholders who are connected by a common approach whose essence is to cultivate the agricultural lands of their ancestors and to revitalize their historical heritage. One of the Croatian examples is the Olive Oil Museum (Museum of olive oil 2022) on the island of Brač. In addition to sightseeing, the tourists can participate in working on the olive groves, as well as tasting and buying local food. The following innovation is the plant nursery for the indigenous species *Anemona*, being an example of traditional seedling production indigenous to the area in accordance with the HNV practice. By researching and monitoring the demand, farmers produce vegetables, flower seedlings, and other planting material for the local market. They are part of a local partners network and institution: they cooperate with schools with regard to the educational activities and with a local radio station with regard to promotional activities. As a garden center, they participate in education and raise the islanders' awareness about local rural areas of a high nature value (HNV). They participate in the preservation of biodiversity and complement the tourist services of Korčula Island with their offers and sales. According to Roglic (2017), what was stated earlier, the realization of innovations in the long term depends on the provision of financial resources for the costs of implementing the action plans, education, and cooperation in the areas of implementation and product branding. Change management processes were not designed to achieve a high nature value (HNV) or conservation goals; however, it is likely that this will be achievable in the long term through the results of maintaining extensive grazing systems and reducing the share of scrub. Potentially, change management can be adapted to special HNV goals through the involvement of competent environmental protection institutions.

### 3.4. Indicators and Typologies of Social Innovations in the Peripheral Rural Areas

Although there is an increasing number of scholars who are researching social innovations at present, the demands for categorizing the field are increasing (Rüede and Lurtz 2012; Jaeger-Erben et al. 2015). According to the research results, the following structure is formed.

The results presented in Table 2 show the structure of indicators and types of social innovations systems.

**Table 2.** The indicators and typologies of social innovation system determinations.

| Types | | Indicators | NESI | | | WESI | | | CESI | | |
|---|---|---|---|---|---|---|---|---|---|---|---|
| | | | NESI[1] | NESI[2] | NESI[3] | WESI[4] | WESI[5] | WESI[6] | CESI[7] | CESI[8] | CESI[9] |
| Social impact | 1 | Personal responsibility | ✓ | X | ✓ | ✓ | ✓ | ✓ | ✓ | X | ✓ |
| | 2 | Social security | X | ✓ | ✓ | ✓ | X | X | X | ✓ | ✓ |
| | 3 | Poverty alleviation | X | X | X | ✓ | X | ✓ | ✓ | ✓ | X |
| | 4 | Social partnerships | ✓ | ✓ | ✓ | X | ✓ | ✓ | ✓ | X | X |
| | 5 | Personal self-sustainability | ✓ | X | X | X | ✓ | ✓ | ✓ | ✓ | ✓ |

**Table 2.** *Cont.*

| Types | | Indicators | NESI | | | WESI | | | CESI | | |
|---|---|---|---|---|---|---|---|---|---|---|---|
| | | | NESI[1] | NESI[2] | NESI[3] | WESI[4] | WESI[5] | WESI[6] | CESI[7] | CESI[8] | CESI[9] |
| Environmental impact | 1 | Habitat protection | ✓ | ✓ | ✓ | ✓ | X | X | ✓ | ✓ | ✓ |
| | 2 | Organic production | ✓ | X | X | X | X | X | X | ✓ | X |
| | 3 | Ecological orientation | ✓ | X | ✓ | X | ✓ | ✓ | X | X | X |
| | 4 | Biodiversity conservation | ✓ | ✓ | ✓ | ✓ | ✓ | X | ✓ | ✓ | ✓ |
| | 5 | Carbon actions and agroforestry | X | ✓ | X | ✓ | X | X | X | X | X |
| Economic impact | 1 | Farm profitability | ✓ | ✓ | X | X | ✓ | ✓ | X | ✓ | X |
| | 2 | Competitive growth | ✓ | ✓ | ✓ | X | ✓ | ✓ | X | ✓ | X |
| | 3 | State support | X | X | X | ✓ | X | X | ✓ | X | ✓ |
| | 4 | Connection of micro and macro actors | X | ✓ | X | ✓ | ✓ | ✓ | ✓ | ✓ | ✓ |
| | 5 | Public investments | X | ✓ | X | ✓ | X | X | X | X | ✓ |

Based on all the recognized occurrences of indicators, it can be determined that the most represented in the structure category are social impacts (37.33%) in the entire research geographical area. In second place, according to the indicator structure, is economic impacts (32.00%), while environmental impacts have the smallest share (30.67%). For the purposes of a graphic representation, all recognized social impacts are summarized (five indicators, one to nine case studies), and then the social impact shares are calculated for each geographical zone. In the same way, the results pertaining to environmental and economic impacts were obtained by summarizing and determining the share.

The results of the comparison of the social, environmental, and economic types (Figure 2) confirm the differences existing within social innovations systems.

The environmental type has a dominant place (43.48%), followed almost equally by social (28.57%) and economic (27.95%) types in the NESI-based social innovation system. The results for WESI show the largest share of economic type (37.50%), this is followed by social (35.71%) and the smallest one is the environmental type (26.79%). The CESI-based social innovation system shows that the highest column type is social (35.71%), the slightly lower column is economic (33.33%), and this is followed by the environmental type (30.95%). The types of impacts of social innovations are different from one country to another. The influences can be stronger or weaker, which assumes that social innovations will be created in completely different ways. The influences of complementary institutions (public or private) to strengthen the target effects also change practices that are recognizable in the structure of types of social innovations (NESI, WESI, and CESI).

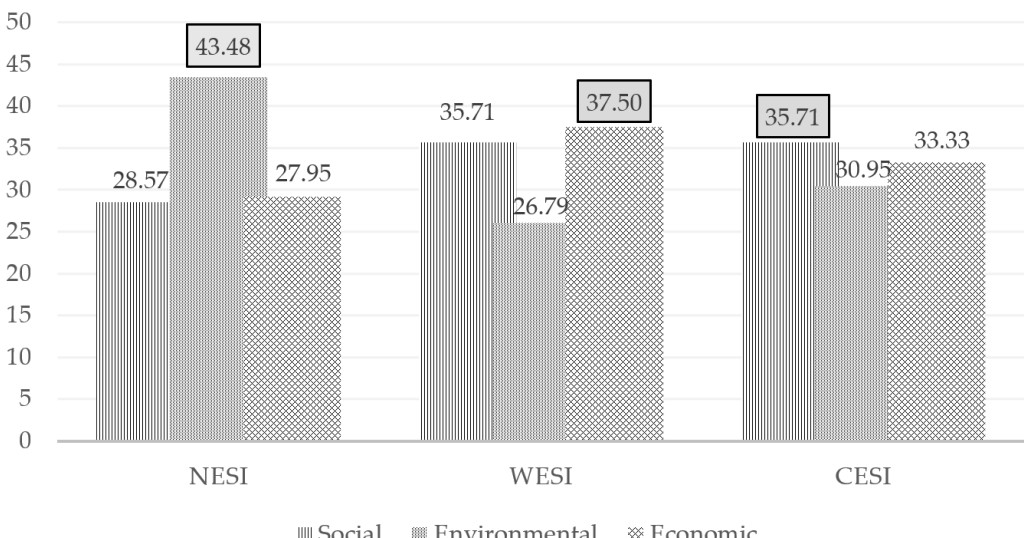

**Figure 2.** A comparison of social innovations systems: social, environmental, and economic types (%).

## 4. Discussion

In many European countries, rural development sustainability practice is directly related to agriculture as a primary economic activity, especially when it comes to peripheral rural areas. The strengthening of agriculture has a synergistic effect on both the processing and the food sectors, as well as on the service tourism industry. The innovations in the peripheral rural areas, where economic activities are directly or indirectly linked to agriculture, become a catalyst for sustainability, adaptation, and social change. Managing the innovation implementation process is an extremely important process because the innovations themselves do not mean much if there is no inventiveness or consistency in their practical applications. Our key findings highlight the differences between the analyzed areas with their specificities in social innovation systems via the following research question:

RQ1: What are the environmental impacts on a social innovation system in the PRAs?

In the NESI-based social innovation system in the PRAs, the environmental type of social innovation dominates when compared to the other types (Figure 2). Based on the collection of knowledge obtained from the analyzed cases, the facts were used as the indicators for a classification according to the types. In the environmental types of social innovation systems, the following indicators were used: habitat protection (3/3), organic production (1/3), environmental orientation (2/3), biodiversity conservation (3/3), and carbon action and agroforestry (1/3), as illustrated in Table 2. The results of a comparative analysis prove that a NESI-based social innovation system has a dominant orientation toward the realization of activities in the context of an environmental type by habitat protection (3/3) and biodiversity conservation (3/3).

An orientation toward the indicators monitored recognized the importance of agricultural lands featuring an HNV. Biodiversity is precisely related to agricultural landscapes and habitat protection where the environmental conditions and sustainability are maintained through low-intensity and low-productivity production practices. Under the influence food market globalization, agroenvironmental production and HNV systems are increasingly disappearing. The answer to the devastation processes, however, was discovered while directing the development of the agricultural systems of a high nature value and instituting habitat protection based on the strength of local resources. In this way, the innovations were offered to local actors via intermediaries in order to preserve the environmental characteristics of agricultural areas of a high nature value, while at the same time improving socioeconomic sustainability. Figure 2 shows how an environmental impact assumes the

role of a leader and how it is almost equally followed by the social and economic impacts of social innovation.

RQ2: What are the economic impacts on a social innovation system in the PRAs?

The focus on economic types prevails in WESI-based social innovation systems, as confirmed by a comparative analysis (Figure 2). In this type of social innovation system, the following indicators were used: farm profitability (2/3), competitive growth (2/3), state support (1/3), connection of micro and macro actors (3/3), and public investments (1/3). The comparative analysis showed that the WESI-based social innovation system had a dominant orientation toward the realization of activities in the context of economic type by virtue of its connection of micro and macro actors (3/3), as illustrated in Table 2.

On the local level, numerous communication activities were conducted in order to raise awareness of the specifics of an area and its potential for agricultural production. Based on a participatory approach, a sustainable development strategy was created that included all actors and priorities: the aggregation of local stakeholders, an exchange of experiences, and then the promotion of needs and proposals for the sake of local community development. A special focus was placed on the land management for pasture conservation, for example, the growers created official signs that identified the quality and origin of brands existent in a region. Platforms were created as well, which brought together the local product suppliers on the one hand (i.e., micro actors), and hand caterers and other institutions (i.e., macro actors) on the other hand. Through connection and cooperation, they ensured the delivery of local products in a simple, direct, and rapid manner. The development of direct distribution improved the producers' economic autonomy. The economic impacts, on the other hand, are visible in the research results (Figure 2) by virtue of a systematic transformation directed toward innovations, with an objective of sustainable development.

RQ3: What are the social impacts on a social innovation system in the PRAs?

In the CESI-based social innovation system in the PRAs, the social type dominates if compared to the other types (Figure 2). In a search for an answer to this research question, the following indicators were used: personal responsibility (2/3), social security (2/3), poverty alleviation (2/3), social partnership (1/3), and personal self-sustainability (3/3). In the CESI-based social innovation system, however, dominant was an orientation toward the realization of activities in the context of personal self-sustainability (3/3), as illustrated in Table 2.

In Croatia, the basics of an economic activity were oriented toward the agriculture and tourism industries, which contributed to personal self-sustainability while employing local and other available resources. An achievement which produced a social impact factor in a tested area, as a peripheral rural zone, was tourism industry (especially continental tourism), with a purpose to exert an effect on unemployment, demographic challenges, poverty, environment, and education. Therefore, the action of civil society organizations was identified as one of the methods that contributed to social and economic development. One of their activities was education for sustainable development. Croatia has a growing community of SMEs as the basis for building effective social innovations. The number of associations, cooperatives, and civil society organizations is increasing due to the support provided by the individuals employed as social workers and those working in the humanities sector who contributed to the improvement of environmental awareness. Innovation processes are limited by the lack of an agricultural data ecosystem, which is the responsibility of stakeholders: agricultural producers/farmers, suppliers, consumer organizations/consumers, public support institutions and researchers, and academia.

RQ4: What are the differences in a social innovations system among the European countries?

By mapping the main forms of social innovation applications, the differences in European countries were identified. The differences in the social innovation system arise from the potential of human resources and their focus on the realization of strategic goals. When strategic actors are committed to the preservation of an area, they act toward the



preservation of biodiversity and agricultural development practices. The question is how to involve other strategic actors who are not primarily focused on environmental sustainability in the implementation of social innovations. The process of their engagement must be carefully considered because it is directly related to the territorially comparative advantages of a rural area. In this sense, the intention of our investigation was to observe the differences evident and contribute to the comprehension of the implementation of social innovation types. Practical approaches to the development of peripheral rural areas suggest numerous innovative solutions; however, the priorities of biodiversity conservation for the purpose of agricultural development can rarely be observed, which consequently disperses the social innovation types. At the same time, the innovations committed to biodiversity conservation and profitable, sustainable agricultural production are suddenly at stake.

The question is: what will prevail? By questioning the nature of practical innovations, stakeholders have the upper hand according to their social, professional, scientific, and financial strengths. A real challenge is to apply social innovation in such a way to achieve rapid economic effects—that is, profitability—along with a long-term strategic reflection on the preservation of favorable environmental conditions, while simultaneously strategically facing the extremely demanding challenges of the social contexts of self-sustainability, employment, and depopulation.

## 5. Conclusions

The expectations concerning social innovation impacts are growing, which simultaneously increases the responsibility for their implementations. An alteration process in sustainable agriculture and the development of peripheral rural areas are connected to the new challenges in the context of social, environmental, and economic transformations. Previous research is strictly linked to the processes of the application of social innovations related to problem-solving practices in the agriculture sector and peripheral rural areas. The theory frequently connects the term "social innovations" with a sustaining innovation practice and sustainable agriculture as a prime activity conducted in rural areas. A comparative case analysis of the relationship between the impacts of social innovation on sustainable development identified the indicators that represent the groups of impact in the implementation of sustainability practices. The indicators were grouped according to the three sustainable development pillars and represented the impacts on the development of high nature value in agricultural and peripheral rural areas. In a follow-up study based on the groups of indicators, the economic, environmental, and social types of social innovation systems were identified.

The main results highlight the differences between the analyzed peripheral rural areas with their social innovation system specificities. It was determined that the NESI-based social innovation system had a larger share of activities that exerted an impact on environmental sustainability, which was realized through a connection between the agriculture and tourism industry. Therefore, the focus placed on an environmental and innovative approach to food production was in line with the largest biodiversity considering the presence of HNV tracts of land, which is especially appreciated by tourists. Good examples of HNV practices are traditional grazing and forage collections that were created on traditional rural habitats located in Finland.

The second social innovation system was the WESI-based one, with its focus placed on the economic impacts on sustainability practices. Throughout history, social innovation systems have been developed under the influence of national welfare programs, with an emphasis placed on the application of various forms of linkage and collaborative management schemes. This kind of development is based on agriculture, a primary economic sector, in the formation of territorial organizations that provide support to agricultural producers. The experiences confirm that the farmers are the main drivers of socioeconomic and rural development practices. Organic agricultural production is encouraged through cooperative organizations in a way that strengthens the alliance of all who can provide support to agriculture on an operational level.

Finally, the CESI-based social innovation system was predominantly oriented toward social impact on the process of sustainable development. In a confrontation with the challenges of climate change, which have a direct impact on agriculture, the foundations of social development were discovered in the field of social entrepreneurship. The key role of social enterprises is innovative and is performed via actual problem-solving practices through the use of available natural resources for a common social benefit. Thus, the entities in sustainable development practices are NGOs, which are frequently supported and cofinanced by a business sector that recognizes the public interests of their activities. In a future perspective, the application of socio-innovative, sustainable microentrepreneurial activities will primarily enable survival, followed by the development of peripheral rural areas. Indeed, social innovations are, therefore, a form of intangible capital in the context of knowledge. The accumulation of such capital depends on numerous factors that encourage or limit it. The effect it ultimately achieves depends on a stimulating entrepreneurial, business, and financial environment.

Our paper's main limitation was, however, a deployment of case studies as secondary data. The limitations were connected with a search for information sources from e-platforms, and the authors used the final results obtained from previous project activities with regard to the methodologies forming the Case Study Bank with full confidence.

Other potential limitations, which are almost always associated with qualitative research, pertain to the authors' subjective assessments when classifying the data into specific groups. Simultaneously, this opens up a space for new research and a scientific debate about the selected criteria (i.e., indicators) and analytical categories (i.e., social, environmental, and economic impacts). Furthermore, there were also limitations concerning the number of countries included in the research. Covering a larger area would contribute to the results, supporting additional arguments with regard to an orientation toward specific types of social innovation. The authors' intention when writing this paper was, nevertheless, to exert an influence on the policymakers who can recognize and use a social innovation potential for the sake of the development of agriculture and rural areas, although we sincerely doubt that they will implement it directly. Nonetheless, the intention was to initiate a public debate and simply exert influence on our awareness about the importance of social innovation both in agricultural and in rural development. Future research should thus investigate the types of social innovations in specific peripheral rural regions. Once the structure of social innovation is determined, it is necessary to focus on the local needs, because it is a direction for the development of specific types of social innovations that could ultimately provide a good service to policymakers.

**Author Contributions:** Conceptualization, J.D. and T.S.; methodology, J.D.; validation, S.T, T.S. and J.D.; formal analysis, J.D.; investigation, S.T.; resources, S.T.; data curation, S.T.; writing—original draft preparation, J.D.; writing—review and editing, T.S.; visualization, T.S.; supervision, J.D.; project administration, S.T.; funding acquisition, S.T. All authors have read and agreed to the published version of the manuscript.

**Funding:** This research received no external funding.

**Informed Consent Statement:** Not applicable.

**Data Availability Statement:** The databases used are publicly available via the following links: https://www.cpdlearnonline.ie/my/ (accessed on 15 September 2022).

**Conflicts of Interest:** The authors declare no conflict of interest. The EC support for forming the databases does not constitute the support of the paper's contents, which only reflect the authorial views, and the Commission cannot be held responsible for the information contained therein. The fund allocators played no role in the design of the study; in the collection, analyses, or interpretation of the data; in the writing of the manuscript; or in the decision to publish the results.

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
