# Peer review of "Social Innovations for the Achievement of Competitive Agriculture and the Sustainable Development of Peripheral Rural Areas"

_economies, doi:10.3390/economies11080209_

Round 1

Reviewer 1 Report

The paper explores the role of social innovation in rural development in achieving sustainable development goals. Through a comprehensive analysis, it identifies and categorizes social innovations in peripheral rural areas. It discusses the challenges of implementing agricultural innovations in rural areas as they affect economic activities and contribute to social and economic sustainability. Overall, the idea of the paper is both interesting and insightful. However, results could be more clearly defined. Therefore, I have the following remarks:

1.    The Introduction section is good. It provides a clear and concise overview of the research topic, highlighting its importance and relevance to the field. I would suggest that the authors further clarify the specific research gap that their study aims to address and the research questions.

2.    In the Methodology section, I noticed that the authors have included some general statements and descriptions (lines 184-193). It is recommended that the authors avoid relying on these generic explanations and provide more specific and detailed information instead.

3.    The authors should provide further clarification on the criteria used to select the 9 cases for their analysis. It would be beneficial to explain the rationale for their choice of specific sites representing peripheral rural areas at the national level of the three selected countries.

4.    The authors present an opinion on the neglect of social innovation and the role of institutions in academic discussions of socio-economic problems (lines 230-232). While their point of view is thought-provoking, it would be beneficial for the authors to support their claim with empirical evidence or references to academic literature.

5.    The authors write that the analysis of European development trends in peripheral rural areas proves that there is enough experience based on which to understand what social innovation is and its importance for development policy (lines 234-236).  It would greatly strengthen their argument if they could support this claim by providing references or citing specific studies that have examined these European development trends and their relationship to social innovation.

6.    Informative tables such as Table 2 and Figure 2 need more explanation. Figure 2 is potentially very important, but it does not seem to be explained at all, while line 431 states that 'There are numerous influences of the external business environment that have implications for the structure of social innovation systems'.

7.    It is important for the authors to address the research gap and thoroughly discuss the research questions in the discussion section of their paper.

8.    It is important for the authors to include a discussion of the limitations of their research and to suggest areas for further investigation in the conclusion section of the paper.

I hope this feedback is helpful, and I wish you the best of luck with your paper.

While the content of the paper is sound, there are some instances where the language could be improved for better clarity and readability. Moderate editing can help to eliminate grammatical errors, improve sentence structure, and make the paper more accessible to readers.

Author Response

Dear reviewer,

I sincerely thank the reviewers for their valuable scientific comments. Your contribution to our manuscript resulted in significant changes and improvements in the structure of the paper, in introduction, research methodology and presentation of the obtained research results. In fact, the current paper was created on the basis of constructive, realistic and above all necessary changes in each segment of the manuscript which is significantly contribute to the paper.

Reviewer 2 Report

This paper addresses a topic of current global interest : social innovation, in a context (rural/agriculture) where innovation is essential in the face of climate variability.

The main problem with the paper is that it is not possible to link the results presented to the sources of data and methodology (interview? with whom? written documents?) nor is there any reassurance that the conclusions follow from the results. Was ethics approval obtained for any interviews? There is a 'black box' that hides exactly what data the researchers gathered, from whom and how it was analysed.

There is discussion of qualitative case study methodology, but Figure 2 presents quantitative date (percentages) comparing social types. The index used to analyse the results is not sufficiently explained.

There may well be a sound research paper hiding in here somewhere, but the authors need to clearly descibe their sample, and draw explicit lines between data sources, results and conclusions.

The paper is in no way suitable for publication in its current form.

Minor typos only.

Author Response

(The authors gave the same response as above.)

Reviewer 3 Report

On the one hand, the topic of the paper is very important and needs to be analyzed and published. On the other hand, the manuscript has several shortages, therefore I have the following comments that can help to improve the scientific value of the paper:

1. Introduction of main approaches to peripheral rural area issues for a better understanding of them as context for sustainable development is missing.

2. Quantitative and qualitative methodology (Chapter Methodology) is explained in a very general way. It must be improved and the application of all used methods for data collection should be described in detail.

3. Contextual data and information about peripheral rural areas in 3 selected countries are missing (e.g. their spatial share, the share of rural population living there, main economic, ecological and social features, etc.). It could help better to understand results and discussion as well as formulation of conclusions.

4. The authors do not point out the limitations of their study in the chapter Conclusions. 

Author Response

(The authors gave the same response as above.)

Round 2

Reviewer 1 Report

Dear Authors,

Thanks for your valuable input.

I found your research to be insightful and thought-provoking. The clarity of your writing and the logical flow of your arguments also made the paper an engaging read.

However, in reading your paper, I came across a few comments and questions that I believe would further enrich the discussion and strengthen the overall impact of your work. I would be very grateful if you could take the time to address these comments:

1. It is highly recommended that the research questions are explained in the introduction section. While the discussion section is important for interpreting and discussing the results of the study, it is not the ideal place to introduce research questions. 

2. The answer to a research question should undoubtedly be based on the author's research and analysis. In addition, the paper should use rigorous analytical techniques to analyse the data or evidence collected. 

3. The paper should provide further clarification of the criteria used to select the 9 cases for analysis. It would be beneficial to explain the rationale for the choice of specific sites representing peripheral rural areas at the national level of the three selected countries.

4. Informative tables such as Table 2 and Figure 2 need more explanation. Figure 2 is potentially very important, but it does not seem to be explained at all.

I believe that addressing these points would enhance the understanding and significance of your research. I look forward to your response and further insights into these issues.

Thank you again for your excellent work. I eagerly await your response to these comments and the opportunity to engage in a fruitful discussion.

I would like to highlight that there is a need for minor editing of the English language in the paper.

Author Response

Dear Reviewer, I sincerely thank for your valuable scientific comments. Your contribution to our manuscript resulted in changes and improvements in the structure of the paper, in introduction, research methodology and presentation of the obtained research results. In fact, the current paper was created on the basis of constructive, realistic and above all necessary changes in each segment of the manuscript witch is significantly contribute to the paper.

Reviewer 2 Report

The revisions have improved the paper. 
While there are still some issues with the structure as an academic research paper (see below) there is one methodological issue that needs to be considered and explained. It appears that determination of whether or not an indicator is present in, or applies to a case is a binary yes/no decision. The presence or absence of indicators such as poverty alleviation or habitat protection could be expected to either be a subjective decision, or be present on a scale from “a little” to “widespread” or similar. There must be explanation of how the binary decisions were made. Was there moderation among a group of researchers? Or did, for example, protection of 10 square metres of habitat in a project area of hundreds so square kilometres earn a case a tick for habitat protection? The  paper must not be published without clear explanation of the process, and discussion of the limitations of that process.

Structural issues  

Line 274 need to note which table and figure numbers, and ideally copy the list of indicators from Table 2 and put them about here.

The actual Results start at Line 319.  Most of the material earlier in the Results section should be moved to the Discussion and interspersed with it as relevant points are discussed. Some may belong in the Background literature section at the start of the paper, especially the 2nd paragraph. 

Minor wording issues
The wording at line 496 needs to talk about ANALYSIS of the cases…

Line 503 not clear what is meant by operation of social innovation, as paper is about types of impact, clarify.

Lines 504-505 makes an implicit assumption that is not explicitly backed up by the findings or the literature, rewording may fix this.

Author Response

(The authors gave the same response as above.)

Round 3

Reviewer 1 Report

Dear Authors,

In the formulation of a research question, it is crucial that it is in the right form. It should be phrased to be objective and unbiased, avoiding leading or loaded language. I would suggest changing the wording for RQ1-3 in the introduction section as it is mentioned in Results.

Table 2 and Figure 2 are presented in Section 3.4. In the Discussion section there is some explanation of the logic of the construction of Table 2 and Figure 2. It would be valuable to have an explanation of the results in Table 2 and Figure 2 in section 3.4.

Author Response

Dear Review, thank you for all your valuable comments, from which I learned a lot.

I am sending the paper as it looks now. Best regards.

Reviewer 2 Report

Revisions have addressed my concerns about lack of clarity re analysis. It is now an excellent paper.

Author Response

(The authors gave the same response as above.)
